# One-Shot Sequential Federated Learning for Non-IID Data by Enhancing Local Model Diversity

Naibo Wang
Institute of Data Science, National
University of Singapore
Singapore, Singapore
naibowang@u.nus.edu

Yuchen Deng
School of Mathematics and Statistics,
Changchun University of Technology
Changchun, China
dengyuchen.cc@gmail.com

Wenjie Feng*
Institute of Data Science, National
University of Singapore
Singapore, Singapore
wenchiehfeng.us@gmail.com

Shichen Fan
School of Computer Science and
Technology, Xidian University
Xi'an, China
shichenfan@stu.xidian.edu.cn

Jianwei Yin
College of Computer Science and
Technology, Zhejiang University
Hangzhou, China
zjuyjw@cs.zju.edu.cn

See-Kiong Ng
Institute of Data Science, National
University of Singapore
Singapore, Singapore
seekiong@nus.edu.sg

## Abstract

Traditional federated learning mainly focuses on parallel settings (PFL), which can suffer significant communication and computation costs. In contrast, one-shot and sequential federated learning (SFL) have emerged as innovative paradigms to alleviate these costs. However, the issue of non-IID (Independent and Identically Distributed) data persists as a significant challenge in one-shot and SFL settings, exacerbated by the restricted communication between clients. In this paper, we improve the one-shot sequential federated learning for non-IID data by proposing a local model diversity-enhancing strategy. Specifically, to leverage the potential of local model diversity for improving model performance, we introduce a local model pool for each client that comprises diverse models generated during local training, and propose two distance measurements to further enhance the model diversity and mitigate the effect of non-IID data. Consequently, our proposed framework can improve the global model performance while maintaining low communication costs. Extensive experiments demonstrate that our method exhibits superior performance to existing one-shot PFL methods and achieves better accuracy compared with state-of-the-art one-shot SFL methods on both label-skew and domain-shift tasks (e.g., 6%+ accuracy improvement on the CIFAR-10 dataset). Our code and supplementary are available online: https://github.com/NaiboWang/FedELMY.

## CCS Concepts

• **Computing methodologies → Distributed algorithms**.

## Keywords

Sequential Federated Learning, One-Shot Federated Learning

---

*Corresponding author.

**ACM Reference Format:**
Naibo Wang, Yuchen Deng, Wenjie Feng, Shichen Fan, Jianwei Yin, and See-Kiong Ng. 2024. One-Shot Sequential Federated Learning for Non-IID Data by Enhancing Local Model Diversity. In *Proceedings of the 32nd ACM International Conference on Multimedia (MM '24), October 28–November 1, 2024, Melbourne, VIC, Australia.* ACM, New York, NY, USA, 10 pages. https://doi.org/10.1145/3664647.3681054

## 1 Introduction

Federated learning (FL) [40] is a promising paradigm which enables collaborative machine learning [2, 50] amongst multiple clients to build a consensus global model without the need to access others' datasets. This paradigm offers salient benefits such as preservation of privacy [17], security of data [39], and the facility for different clients to derive a model exhibiting a higher degree of inference capability compared to individual client-based training [55].

As shown in Fig. 1, two federated learning paradigms are parallel FL (PFL) [5, 41] and sequential FL (SFL) [38]. PFL synchronizes model training across clients in parallel, such as FedAvg [40], while SFL adopts a client-by-client training sequence that is widely applied in many scenarios such as healthcare [8, 9]. Compared with PFL, SFL exhibits significant advantages in training efficiency, as evidenced by a reduction in training rounds [57]. SFL also shows robust performance with limited datasets [28] and offers enhanced data privacy protection due to its decentralized architecture [21].

One main concern of SFL is communication costs. To mitigate communication costs, one-shot federated learning [19] has been proposed where only one communication round is needed for the clients to interact with the server or other clients. However, current one-shot FL works [22, 58] mainly focus on PFL (Fig. 1 (a)) which typically requires a central server to facilitate model training. This can lead to numerous limitations including risk of privacy leakage [27] and server node bottlenecks [20]. To address these issues, the decentralized structure [47] has been incorporated into one-shot PFL which typically utilizes a mesh-topology network for clients to disseminate their models to others. Nonetheless, this setting still leads to significant communication overhead than SFL. In contrast, one-shot SFL, wherein each client only needs to communicate with its adjacent client once, can greatly reduce communication costs. However, existing studies on one-shot SFL [8, 44] struggle with

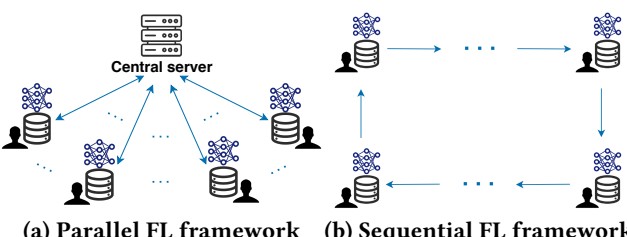

**(a) Parallel FL framework**  **(b) Sequential FL framework**

**Figure 1: Two federated learning settings.**

handling the non-IID data, a common challenge in FL that significantly impairs the performance of federated models [46]. How to tackle the non-IID data in one-shot SFL is still an open problem.

The key question in one-shot SFL is how to effectively transfer knowledge given the limited communication, especially with the non-IID data. Our insight is to utilize the diversity of models. Existing studies have shown that combining multiple neural networks can notably enhance the model's performance due to the inherent diversity amongst model weights [7, 25, 52]. This observation has also been solidly backed by theoretical support from Rame et al. [42]. However, directly applying such a diversity strategy in SFL is not feasible, since SFL restricts each client to receive only one model from its adjacent client . This limitation could lead to insufficient diversity during model training, which can potentially undermine the performance of the final trained model. To overcome this challenge within the one-shot SFL framework, we propose a diversity-enhanced mechanism for model training. This mechanism is designed to augment the diversity of locally trained models. By generating a broader spectrum of models, we facilitate enriched knowledge transfer between adjacent clients, which also serves to mitigate the impact of data distribution disparities across clients.

In this paper, we present a novel one-shot sequential *Fed*erated learning framework by *E*nhancing *L*ocal *M*odel diversit*Y* (FedELMY). Specifically, we improve the local model training diversity by constructing a *model pool* consisting of various models for each client. We introduce two distance regularization terms during the local training process to enhance model training diversity while mitigating the impact of non-IID data. Fig. 2 presents a case study illustrating our core concept. Compared with conventional federated learning methods, our approach effectively minimizes communication costs, safeguards data privacy, and mitigates the detrimental effects of non-IID data, thereby enhancing overall model performance. Experiments show that our method can outperform existing one-shot SFL methods on both label-skew and domain-shift datasets.

The main contributions of this paper are as follows:

- We tackle the novel and practical problem of one-shot sequential federated learning. To the best of our knowledge, this is the first work to systematically investigate the one-shot communication setting in sequential federated learning.
- We introduce a novel framework FedELMY to reduce the communication cost and improve the global model performance by enhancing model diversity during local training.
- We conduct extensive experiments on four datasets, considering both feature and label distribution shifts. Our method

achieves superior performance compared with existing one-shot PFL and SFL methods.

## 2 Related Work

### 2.1 Parallel Federated Learning

One well-known PFL method is FedAvg [40], whose performance is hindered due to the dispersed nature of the data (non-IID). Methods such as FedProx [36], FedDyn [1], Astraea [13], pFedMe [49], and FedCurv [46] use a measure of global parameter stiffness to tackle data heterogeneity in federated learning. Additionally, approaches like FedDC [16] and SCAFFOLD [29] use a global gradient adjustment mechanism to manage local variations of the data. Other methods like FedGMM [53], FCCL [23], FedBN [37] and ADCOL [34] utilize the personalized model instead of a shared global model to improve the performance of federated learning. These methods all necessitate a central server for model training.

Another PFL scenario is the decentralized federated learning [3, 43, 56] when edge devices conduct training without a central server. Sun et al. [48] proposed the decentralized FedAvg with momentum (DFedAvgM) algorithm to improve the performance of trained models in the decentralized federated learning setting. Shi et al. [45] applied the SAM [15] optimizer to improve the model consistency of decentralized federated learning. This setting can better protect privacy than centralized PFL but incurs more communication costs.

### 2.2 Sequential Federated Learning

Recently, Sequential Federated Learning (SFL) started to gain attention in the FL community. Micah J et al. [44] proposed a basic SFL framework to facilitate multi-institutional collaborations without sharing patient data; Li et al. [38] proved that the convergence guarantees of SFL on heterogeneous data are better than PFL for both full and partial client participation, and validated that SFL outperforms PFL on extremely heterogeneous data in cross-device settings; Chen et al. [9] proposed MetaFed, an SFL scheme with cyclic knowledge distillation for personalized healthcare; Cho et al. [10] also provided convergence analysis of sequential federated averaging and proves that it can achieve a faster asymptotic convergence rate than vanilla FedAvg with uniform client participation under suitable conditions. However, existing works still lack the ability to deal with non-IID data and exhibit poor performance in real-world applications.

### 2.3 One-shot Federated Learning

One-shot federated learning [12, 14, 19, 47], which aims to train a global model with only one round of communication between the clients and the server, has been proposed to reduce the communication cost and simultaneously enhance data privacy protection in federated learning. FedDISC [54] made progress in one-shot semi-supervised federated learning, employing a pre-trained diffusion model. Investigations by Diao et al. [11] and Joshi et al. [26] have independently explored one-shot federated learning, viewing the problem through the respective perspectives of the open-set problem and Fisher information. Moreover, the proliferation of pre-trained models has sparked interest in collaborative, model-centric machine learning [4, 51]. Instances such as FedKT [33] and DENSE [58] are two one-shot federated learning schemes that

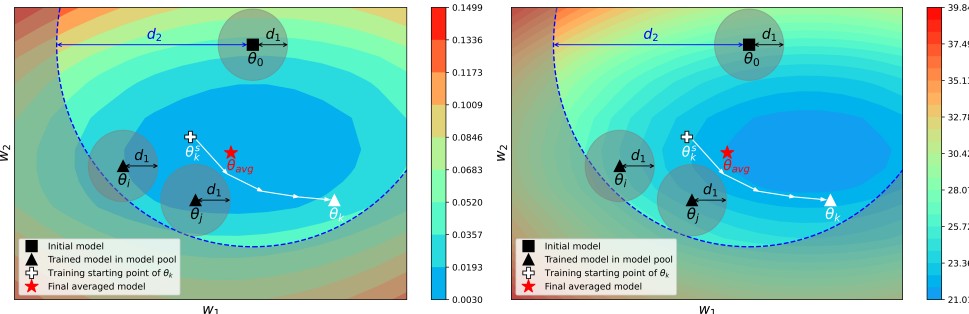

(a) Training loss surface on local dataset $D_i$.            (b) Test error surface on the whole test set.

Figure 2: An illustration of our training solution on client $i$. Based on previously trained models in the model pool, every new model $m_k$ starts training from $\theta_k^s = f(\{\theta_i\}_{i=0}^{k-1})$ to improve training diversity ($f$ is the average function in our paper). During training, optimization of $\theta_k$ is constrained within a specific region (the non-shadow areas). $\theta_k$ is required to maintain a certain distance ($d_1$) from existing models $\{\theta_i\}_{i=0}^{k-1}$ to enhance model diversity, and should not diverge significantly ($d_2$) from the initial model $\theta_0$ to prevent deviation from the globally optimal solution caused by non-IID data. After training, all trained models $\{\theta_i\}_{i=1}^{k}$ display similar training losses on the local dataset $D_i$ (a) but have different test errors on the whole test set (b). Meanwhile, the averaged model $\theta_{avg}$ of all models in the model pool achieves a lower test error than any single model (b).

choose to send models instead of gradients to the central server and generate a final global model through knowledge distillation. Despite their advancements, these one-shot methods are not designed for sequential federated learning, which still results in a higher risk of privacy leakage. Thus, devising a one-shot sequential federated learning framework is still under-explored.

## 3 Method

In this section, we introduce our problem setting and present our proposed algorithm FedELMY. The overview of our method is demonstrated in Fig. 3.

### 3.1 Problem Formulation

Assume there are $N$ different clients (parties), each client has its own private dataset $D_i = \{(x_k, y_k)\}_{k=1}^{n_i}$ with size $n_i$. The principal goal of sequential federated learning is to develop a global model $m$ across the dataset $\mathcal{D} = \{D_i\}_{i=1}^{N}$, by minimizing the error on training data. The optimization objective is formulated as:

$$\min_{m} \sum_{i=1}^{N} \mathbb{E}_{(x,y) \sim D_i} [L(m; x, y)], \tag{1}$$

where $L(m; x, y)$ is the loss function evaluated on the private dataset $D_i$ from client $i$ ($c_i$) with model $m$. The training procedure of sequential federated learning is described as follows:

1. At the start of each training round $r$, the client's training sequence $\{c_{\pi_1}, c_{\pi_2}, \cdots, c_{\pi_N}\}$ is determined by randomly selecting indices $\{\pi_1, \pi_2, \cdots, \pi_N\}$ without replacement from set $\{1, 2, \cdots, N\}$.

2. At the very beginning of training when round $r = 1$, randomly initialize global model $m^{(0)}$.

3. For the $i$-th client ($c_{\pi_i}$) in round $r$, initialize its model $m_{\pi_i,0}^{(r)}$ with the latest model:

$$m_{\pi_i,0}^{(r)} = \begin{cases} m^{(r-1)}, & \text{if } i = 1 \\ m_{\pi_{i-1},E_{local}}^{(r)}, & \text{if } i > 1 \end{cases} \tag{2}$$

where $m^{(r-1)}$ is the global model received from round $r - 1$, and $m_{\pi_{i-1},E_{local}}^{(r)}$ is the model received from the $(i - 1)$-th client ($c_{\pi_{i-1}}$) in round $r$ after it trained its model for $E_{local}$ epochs.

4. Update model $m_{\pi_i,0}^{(r)}$ for $E_{local}$ epochs based on $D_{\pi_i}$ and send $m_{\pi_i,E_{local}}^{(r)}$ to the $(i + 1)$-th client ($c_{\pi_{i+1}}$), if SGD [6] is chosen as the optimizer, then the update can be described by:

$$m_{\pi_i,k+1}^{(r)} = m_{\pi_i,k}^{(r)} - \eta \cdot g_{\pi_i,k}^{(r)} \tag{3}$$

where $m_{\pi_i,k+1}^{(r)}$ denotes the local model of client $\pi_i$ after $k$ local training steps in round $r$, $\eta$ is the learning rate, and $g_{\pi_i,k}^{(r)}$ represents the gradient of the loss function based on $D_{\pi_i}$ at step $k$.

5. At the end of round $r$, i.e., after all clients finished their training, we can get the global model $m^{(r)}$ of round $r$ as:

$$m^{(r)} = m_{\pi_N,E_{local}}^{(r)} \tag{4}$$

For one-shot sequential federated learning, only 1 training round is required, i.e., $r \equiv 1$, thus $m^{(1)}$ will be the final global model.

*Analysis.* Although the communication cost is low, the information exchanged in SFL is limited, especially under the one-shot setting. Therefore, it is a big challenge to aggregate diverse and useful information to get a better global model. In other words, the diversity of distributed data under the one-shot SFL paradigm has not been well utilized. Meanwhile, the non-IID data in SFL increases the risk of models getting stuck in a local minimum, which further limits the enhancement of the generalization ability of the global model [36]. Hence, it is essential to explore methods to increase the diversity during model training and mitigate the impact of non-IID data to enhance the model's performance in one-shot SFL.

### 3.2 Local Model Diversity Enhancement

The authors of SWAD [7] demonstrate that solely minimizing empirical loss in a single model is typically insufficient for achieving

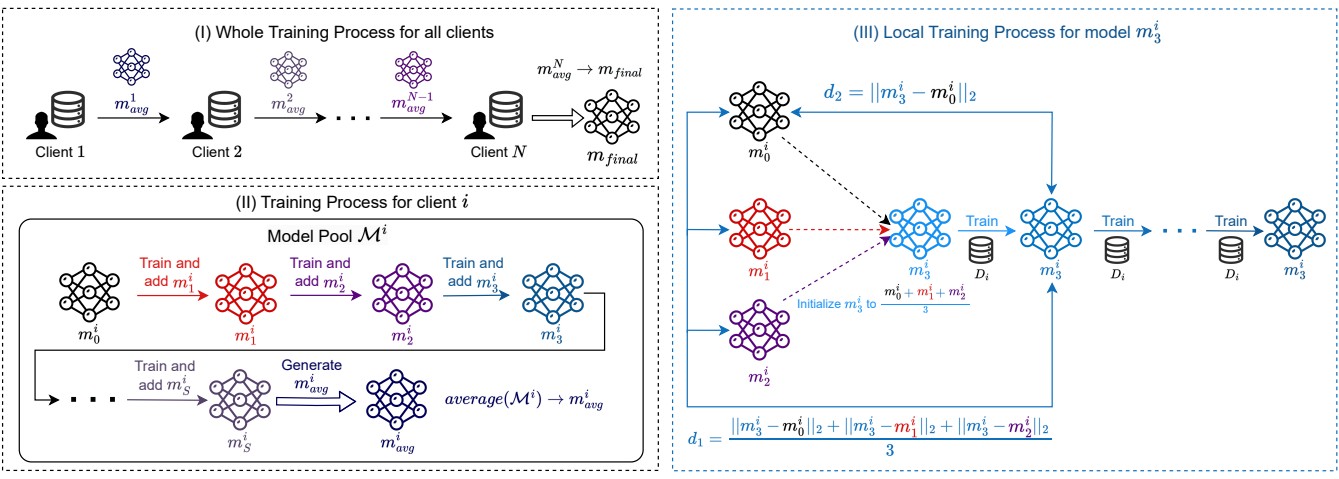

**Figure 3: Overview of our method. Every client $i$ receives a model $m_{avg}^{i-1}$ from its previous client $i-1$ and sends model $m_{avg}^i$ to its next client $i+1$ after training (I). For each client $i$, we train $S$ models and put them into its model pool $\mathcal{M}^i$ (II). Every new model $m_j^i$ is initialized to the average of the existing models in $\mathcal{M}^i$ and trained under the control of $d_1$ and $d_2$ (III).**

good generalization. Furthermore, they argue that the performance markedly improves through averaging diverse models trained by various hyperparameters, even when the training losses of these models are similar. Therefore, we propose to enhance diversity by a *Model Pool*, which is essentially a collection of models maintained by a client during its local training. Here, every client $i$ possesses a model pool $\mathcal{M}^i$ initially consisting of a solitary model, denoted as $m_0^i$. For the first client when $i=1$, the model $m_0^1$ undergoes the process of random initialization, followed by a *warm-up* process over $E_w$ epochs on its local dataset $D_1$. As for the subsequent clients indexed from $i=2:N$, each model, $m_0^i$, is assigned to be the average model $m_{avg}^{i-1}$ which is efficiently derived from the model pool $\mathcal{M}^{i-1}$ of the previous client $i-1$:

$$m_{avg}^{i-1} \leftarrow \frac{1}{|\mathcal{M}^{i-1}|} \sum_{t=0}^{|\mathcal{M}^{i-1}|-1} m_t^{i-1}, \qquad (5)$$

The parameters to be sent to the next client have been improved to an average of a local diverse model pool instead of a single model.

Then, for each individual client $i$, we train an additional series of $S$ models, denoted as $\{m_j^i\}_{j=1}^S$, on the local dataset $D_i$ to probe a broader range of diverse models. Each new model, $m_j^i$, stems from the existing models in the established model pool, $\mathcal{M}^i$. More specifically, initialization of every new model $m_j^i$ is achieved by averaging all the models currently in circulation within the model pool $\mathcal{M}^i$, with the first model $m_0^i$ included; that is,

$$m_j^i \leftarrow \frac{1}{|\mathcal{M}^i|} \sum_{t=0}^{|\mathcal{M}^i|-1} m_t^i, \qquad (6)$$

In this way, we craft a unique launching point for the training of each new model, ensuring a departure from all earlier models within our pool $\mathcal{M}^i$. Consequently, a diverse array of starting points is harnessed, granting us access to a more comprehensive set of potential solutions that evade the restrictions of a singular training trajectory and its attendant limitations.

After initialization, the model $m_j^i$ is then trained for $E_{local}$ epochs, employing the same hyperparameters as with preceding models. Upon finishing training, $m_j^i$ will be added to the model pool $\mathcal{M}^i$. After training all $S$ models for client $i$, the client will calculate the final averaged model $m_{avg}^i$, which is the average of all models in $\mathcal{M}^i$, and send $m_{avg}^i$ to the subsequent client, $i+1$. This process will be repeated until client $i=N$ concludes its local training. Upon completion, the final client $i=N$ will output the final model $m_{final}$, which is the average of all models in its model pool $\mathcal{M}^N$, i.e., $m_{final} \leftarrow \frac{1}{|\mathcal{M}^N|} \sum_{t=0}^{|\mathcal{M}^N|-1} m_t^N$.

To further enhance training diversity and prevent the convergence of the newly generated model $m_j^i$ towards the states of existing models in $\mathcal{M}^i$—thereby diminishing diversity—we aim to maintain the current exploration direction of the model as distinct as possible from those of existing models, throughout the training process. Consequently, we introduce a new distance control term, $d_1$, in the training process, which is defined as:

$$d_1 = \frac{1}{|\mathcal{M}^i|} \sum_{t=0}^{|\mathcal{M}^i|-1} dist(m_j^i, m_t^i) \qquad (7)$$

where $m_j^i$ represents the current model being trained for client $i$, and $m_t^i$ signifies a model in the model pool $\mathcal{M}^i$. We use the $\ell_2$ norm for $dist$, that is, calculating the averaging the $L_2 - norm$ between the current model and all existing models in the model pool. The loss function incorporates and deducts this distance during training, allowing the model to maximize its distance from the existing models, therefore promoting training diversity.

### 3.3 Mitigation of Non-IID Data Impact

In non-IID scenarios, variations in local objectives may risk deviating from the globally optimal solution across multiple local iterations, thereby impeding convergence [36]. To prevent significant deviation of the model from the global solution, we further

**Algorithm 1:** Our proposed FedELMY

**Input:** Local datasets $\mathcal{D} = \{D_i\}_{i=1}^{N}$, warm-up epoch $E_w$, learning rate $\eta$, number of local iterations $E_{local}$, model number to be trained per client $S$, scale hyperparameters $\alpha$, $\beta$

**Output:** The final model $m_{final}$

1 **Initialization:** For client 1, warm up a randomly initialized model $m_{avg}^0$ for $E_w$ epochs

2 **for** *client* $i = 1 : N$ **do**

3      Only for client $i = 2 : N$, receive $m_{avg}^{i-1}$ from client $i-1$

4      // Initialize model pool $\mathcal{M}^i$ for client $i$

5      $\mathcal{M}^i = \{m_0^i\}$ with $m_0^i \leftarrow m_{avg}^{i-1}$

6      **for** $j = 1 : S$ **do**

7          // Initialize $m_j^i$

8          $m_j^i \leftarrow \frac{1}{|\mathcal{M}^i|} \sum_{t=0}^{|\mathcal{M}^i|-1} m_t^i$

9          // Local training for $m_j^i$

10          **for** $k = 1 : E_{local}$ **do**

11              $\mathcal{L}(m_j^i) = \ell(m_j^i; D_i) - \alpha \cdot d_1 + \beta \cdot d_2$

12              $m_j^i \leftarrow m_j^i - \eta \nabla_m \mathcal{L}(m_j^i)$

13          **end**

14          $\mathcal{M}^i \leftarrow \mathcal{M}^i \cup \{m_j^i\}$

15      **end**

16      $m_{avg}^i \leftarrow \frac{1}{|\mathcal{M}^i|} \sum_{t=0}^{|\mathcal{M}^i|-1} m_t^i$

17      Only for client $i = 1 : N - 1$, send $m_{avg}^i$ to client $i + 1$

18 **end**

19 // For the final client $i = N$, output model $m_{final}$

20 $m_{final} \leftarrow m_{avg}^N$

---

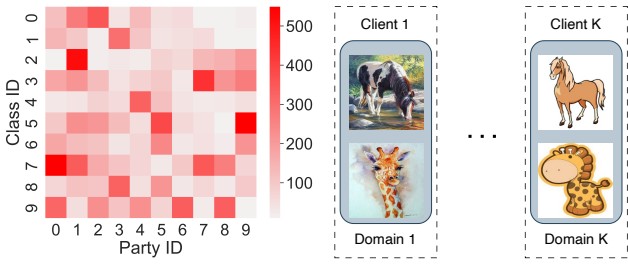

**(a) Label-skew distribution.**    **(b) Domain-shift distribution.**

**Figure 4: Two data distributions across clients. For the label-skew distribution, the color depth of every square represents the number of samples of the corresponding class on that client; for the domain-shift (feature-skew) distribution, every client possesses a specific domain with all classes.**

create and initialize a model pool $\mathcal{M}^i$ (lines 3-5). The process of involving a new model $m_j^i$ in $\mathcal{M}^i$ is presented in line 8. Lines 10-13 elaborate on the local training process per model based on our customized loss function $\mathcal{L}$, as delineated in Sec. 3.3. Once all models for client $i$ have completed training, the average model $m_{avg}^i$ derived from $\mathcal{M}^i$ is sent to the next client $i + 1$ (lines 16-17). The final global model $m_{final}$ is obtained as the average of the models in model pool $\mathcal{M}^N$ of the final client $N$ (line 20).

**Communication cost.** In our method, each client will send a model to its adjacent client only once, thus only $N - 1$ model exchanges are required. Therefore, the overall communication cost for all $N$ parties is $O(NM)$, where $M$ represents the size of the model.

**Computation cost.** In our approach, every client will train $S$ models, thus the overall computation cost for all $N$ parties is $O(NSE_{local})$.

## 4 Experiments

### 4.1 Experimental Setup

**Datasets and Data Partition.** As shown in Fig. 4, in federated learning applications, non-IID data distribution is a common scenario, often arising in two forms: *label-skew* [11] and *feature-skew* (also known as *domain-shift*) [24]. We seek to corroborate the effectiveness of our method under these two distinctive non-IID conditions. We select two datasets associated with label-skew setting, namely **CIFAR-10** [30] and **Tiny-ImageNet** [31], along with two datasets linked to domain-shift setting, specifically **PACS** [32] and **Office-Caltech-10** [18].

The training datasets for CIFAR-10 and Tiny-ImageNet are partitioned into $N = 10$ clients with Dirichlet distribution $Dir(0.5)$. Meanwhile, with respect to the PACS and Office-Caltech-10 datasets, they intrinsically contain four distinct domain samples, each allocated to a single client, yielding $N = 4$ clients in total. Each client is provided with a randomly selected 90% of the local training dataset for training purposes, with the remaining serving as the validation set. We test the final model $m_{final}$ on all test data from all clients, providing a global test performance measurement.

**Baselines.** We compare our method with baselines under various federated learning settings: **DFedAvgM** [48], a decentralized adaptation of the extensively studied FedAvg [40] method in federated

---

refine the model training process by introducing an additional regularization term $d_2$, defined as follows:

$$d_2 = dist(m_j^i, m_0^i) = ||m_j^i - m_0^i||_2, \quad (8)$$

where $m_0^i$ represents the first model in the model pool $\mathcal{M}^i$ for client $i$. The loss function also incorporates this term to ensure that the model maintains reasonable proximity to the initial model $m_0^i$ (global solution of previous clients) in the pool during local updates. It helps mitigate the effect of non-IID data distribution while accommodating system heterogeneity.

The total loss function $\mathcal{L}$ for model $m_j^i$ is formed as follows:

$$\mathcal{L}(m_j^i) = \ell(m_j^i; D_i) - \alpha \cdot d_1 + \beta \cdot d_2, \quad (9)$$

where $\ell$ denotes the original loss function, $D_i$ is the local dataset for client $i$, and $\alpha$ and $\beta$ are two hyper-parameters that govern the effect of both distances on model training. Our aim is to achieve a balance that enhances model training diversity while mitigating the impact of non-IID data.

## 3.4 Implementation Details

The details of our method are depicted in Algorithm 1. The training procedure begins at client 1, which acts as the starting point for training with a warm-up phase (line 1). Then, each client $i$ will

**Table 1: Test accuracy (%, mean±std) comparison of our FedELMY method to other baselines on both label-skew and domain-shift (feature-skew) tasks. MetaFed, FedSeq and our FedELMY are SFL methods, while other baselines are all PFL methods.**

| Distribution | Label-Skew | | | | Domain-Shift | | | |
|---|---|---|---|---|---|---|---|---|
| Dataset | CIFAR-10 | | Tiny-Imagenet | | PACS | | Office-Caltech-10 | |
| $E_{local}$ | 100 | 200 | 100 | 200 | 100 | 200 | 100 | 200 |
| DFedAvgM | 19.32±2.13 | 18.59±1.65 | 2.72±0.52 | 2.02±0.20 | 20.49±1.07 | 21.58±2.28 | 10.18±1.56 | 10.01±0.70 |
| DFedSAM | 17.12±0.44 | 18.51±1.28 | 2.58±0.29 | 3.15±0.29 | 19.01±1.09 | 20.79±1.36 | 14.74±1.63 | 15.09±1.04 |
| FedOV | 36.32±7.56 | 38.06±7.40 | 1.18±0.10 | 1.29±0.30 | 12.09±4.00 | 22.15±1.40 | 9.67±4.09 | 10.01±3.72 |
| DENSE | 61.76±0.43 | 64.59±1.04 | 1.27±0.45 | 1.49±0.05 | 17.34±2.14 | 15.60±5.40 | 33.12±5.16 | 36.68±4.87 |
| MetaFed | 71.46±0.95 | 71.31±0.73 | 23.52±0.29 | 24.76±0.24 | 35.67±0.27 | 38.73±2.61 | 41.78±3.24 | 42.26±3.77 |
| FedSeq | 72.92±0.62 | 73.59±0.95 | 25.50±0.34 | 25.08±0.89 | 43.68±1.29 | 46.53±0.75 | 32.78±3.99 | 37.07±5.78 |
| FedELMY | **79.03±0.74** | **80.28±1.18** | **32.84±0.22** | **30.42±0.10** | **46.08±1.82** | **47.74±1.68** | **44.64±1.98** | **45.26±3.93** |

learning; **DFedSAM** [45], an application of the SAM [15] optimizer in decentralized PFL; **FedOV** [35], a centralized one-shot federated learning method employing the open-set to bolster the final model; **DENSE** [58], a one-shot federated learning framework also reliant on a central server to deliver a global model via knowledge distillation. Lastly, we compare our method to two sequential federated learning methods: **MetaFed** [9], a personalized sequential federated learning approach used for healthcare, which requires at least two rounds of communication; **FedSeq** [38], which is the state-of-the-art sequential federated learning algorithm.

In an effort to ensure a fair comparison, we adapted the decentralized PFL methods DFedAvgM and DFedSAM to the one-shot setting, and adjusted these methods to select all clients for training and communication to fit the setting. In the remaining parts, unless otherwise specified, we set the local training epoch $E_{local}$ to 200. For more implementation details about our method and all baseline methods, such as the optimizer, learning rate and batch size we employed, please refer to Sec. 1 of the supplementary material.

## 4.2 Effectiveness

*4.2.1 One-shot Setting.* Table 1 presents the test accuracy of different methods in two training settings ($E_{local}$ = 100 and 200) to verify the robustness of our method. Results show that our FedELMY method outperforms all other methods across all datasets including both label-skew and domain-shift tasks. Specifically, FedELMY surpasses existing SFL methods by over 6% in accuracy, and outperforms current PFL methods by more than 15% on the CIFAR-10 (label-skew) dataset. Additionally, FedELMY achieves a 25% higher accuracy over PFL methods and at least an 1.5% improvement over SFL methods on the PACS (domain-shift) dataset. These results consistently demonstrate the effectiveness of our method.

It is noteworthy that for the Tiny-ImageNet dataset, due to its large number of classes (200), most PFL baselines fail to deliver effective performance such as DENSE (close to random guesses), suggesting that they cannot effectively handle datasets with too many classes. However, our method works well and also achieves improvement compared to other SFL methods. Such improvements are attributed to the diverse local training so that our method can learn better feature representations.

In addition, it is not always the case that the test accuracy of $E_{local}$ = 200 is better than 100, indicating that simply increasing

**Table 2: Few-Shot performance comparison for PACS dataset.**

| Shot | 1 | 3 | 5 | 7 |
|---|---|---|---|---|
| MetaFed | 41.62% | 44.75% | 46.11% | 46.64% |
| FedSeq | 47.31% | 49.30% | 50.96% | 50.97% |
| FedELMY | **49.14%** | **56.35%** | **57.05%** | **57.13%** |

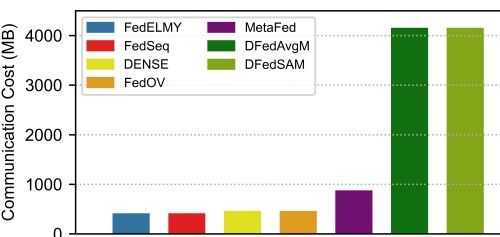

**Figure 5: Communication cost comparison of different algorithms for CIFAR-10 dataset when the number of clients $N = 10$ with model ResNet-18.**

the number of training rounds does not always improve model performance – enhancing diversity is an essential factor.

*4.2.2 Few-Shot Setting.* Although FedELMY is designed for the one-shot setting, here we explore the performance of FedELMY in few-shot scenarios to validate the scalability of our framework, i.e., when the final client $N$ sends model $m_{avg}^N$ to the first client 1, thereby starting a new cycle of model training. As we can see from Table 2, even under few-shot settings, FedELMY consistently outperforms the SFL baselines. Such observations underscore the effectiveness and scalability of FedELMY. Meanwhile, we observe that as the number of training rounds increases to a certain extent, the overall performance does not significantly improve, indicating that the model has reached a state of convergence. This concludes that blindly increasing the number of training rounds will not significantly enhance the model performance.

## 4.3 Efficiency

In this section, we will evaluate our method's efficiency by comparing its communication cost and model performance under different computation costs with other baselines.

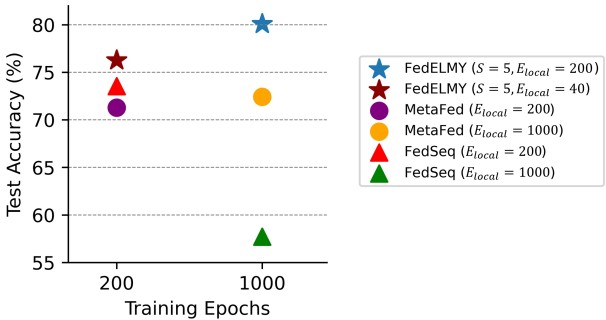

**Figure 6: Test accuracy comparison with different computation costs on CIFAR-10 dataset, where FedELMY ($S = 5, E_{local} = 40$) means we will train 5 models on every client, wherein each model undergoes 40 training epochs in our framework; FedSeq ($E_{local} = 1000$) denotes that we will train only 1 model for 1000 epochs for every client in FedSeq.**

*4.3.1 Communication cost.* Fig. 5 shows the communication expenses of different methods with $N = 10$ clients on the CIFAR-10 dataset with the Resnet-18 model, whose size is $M = 46.2$ MB. The communication cost of FedELMY is restricted to $(N - 1) \times M = 415.8$ MB, with only the FedSeq method displaying the same cost as ours; however, it notably underperforms in performance (Table. 1). MetaFed requires $(2N - 1) \times M = 877.8$ MB of communication cost since it requires at least two rounds of communication (common knowledge accumulation and personalization). Central server-dependent methods, such as DENSE and FedOV, require an expenditure of $N \times M = 462$ MB to transfer models to the server. Other methods mandating communication with their neighbors like DFedAvgM and DFedSAM, will require even higher communication costs. Accordingly, our method effectively diminishes the communication burden of FL and thus strengthens data privacy.

*4.3.2 Computation cost.* Given that FedELMY requires training $S$ models at each client, it involves $S \times E_{local}$ epochs of training per client. Compared to FedSeq, in which each client trains a single model for $E_{local}$ epochs, and MetaFed, which necessitates $2 \times E_{local}$ epochs of training per client, our method appears to demand a higher computational expense. However, FedELMY significantly enhances the model performance without increasing the communication cost (Table. 1). Thus, our method strikes a balance (trade-off) between computational cost (training time) and performance gains.

Meanwhile, we conducted experiments to align the computational costs of different methods for a fair comparison (Fig. 6). First, we align the computation cost of our method to the baselines. By setting $\{S = 5, E_{local} = 40\}$, FedELMY maintains the same computational cost as baselines. As we can see, even under this setting, our method still outperforms the baselines; then, we align the computation cost of the baselines to FedELMY by increasing $E_{local}$ to 1000, Fig. 6 shows that FedSeq displays worse performance and MetaFed shows negligible variation in its performance. This can be attributed to the phenomenon of overfitting due to excessive local training, which increases the model's generalization error and hence, compromises their performance. Contrastingly, our approach, which

**Table 3: Ablation studies on different regularization terms.**

| Method | $\mathcal{M}$ | $d1$ | $d2$ | CIFAR10 | PACS |
|---|---|---|---|---|---|
| MetaFed | | | | 71.29% | 41.62% |
| FedSeq | | | | 73.54% | 47.31% |
| FedELMY | ✓ | | | 78.92% | 48.11% |
| FedELMY | ✓ | ✓ | | 79.62% | 49.04% |
| FedELMY | ✓ | | ✓ | 79.77% | 48.94% |
| FedELMY | ✓ | ✓ | ✓ | **80.08%** | **49.14%** |

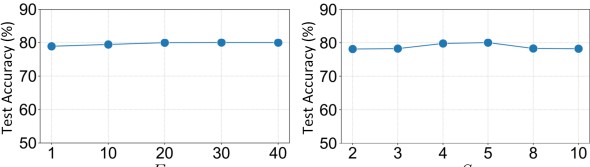

**(a) Effect of warm-up epoch $E_w$. (b) Effect of model quantity $S$.**

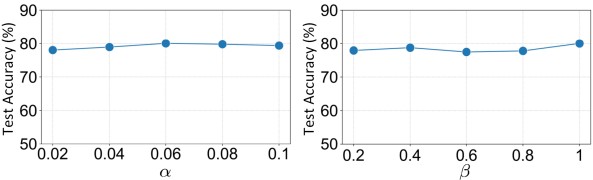

**(c) Effect of scale parameter $\alpha$. (d) Effect of scale parameter $\beta$.**

**Figure 7: Grid search results for CIFAR-10 dataset to investigate the sensitivity of FedELMY to various hyperparameters.**

equally included $S \times E_{local}$ training rounds, exhibited superior performance. These experiments all validate the efficacy of our model training procedures by the diversity-enhanced mechanism.

## 4.4 Ablation study

In this section, we examine the impact of distance metrics, hyperparameters, and diversity control measures on our method's performance, as well as the influence of client orders.

*4.4.1 Effects of distance terms.* We conducted ablation experiments to verify the effectiveness of the components in our method. Since the model pool $\mathcal{M}$ is indispensable, we examined the impact of the two distance regularization terms, $d_1$ and $d_2$, referenced in Eq. (7) and Eq. (8) respectively. Table 3 illustrates the outcomes of respective schemes. As we can see, the introduction of either $d_1$ or $d_2$ independently improves our method's performance compared to solely using the model pool $\mathcal{M}$, and further enhancement is achieved when both distance terms are incorporated. This confirms that integrating both distance terms will strengthen the model's performance. It is worth emphasizing that even in the absence of $d_1$ and $d_2$, relying solely on the model pool $\mathcal{M}$ for training, our approach still outperforms the baselines. This reaffirms the essential paradigm for fostering diversity within the system.

*4.4.2 Hyperparameter sensitivity.* We further investigate the sensitivity of our method to different choices of hyperparameters, as

**Table 4: Performance comparison of PACS dataset for different domain training orders, where the order "PACS" means we train models by domain order "*Photo* (client 1) → *Art-Painting* (client 2) → *Cartoon* (client 3) → *Sketch* (client 4)".**

| Order | PACS | ACPS | SCPA | CSPA | Average |
|---|---|---|---|---|---|
| MetaFed | 41.62% | 42.75% | 31.65% | 40.64% | 39.17% |
| FedSeq | 47.31% | 45.21% | 40.96% | 33.71% | 41.80% |
| FedELMY | **49.14%** | **46.74%** | **43.65%** | **41.46%** | **45.25%** |

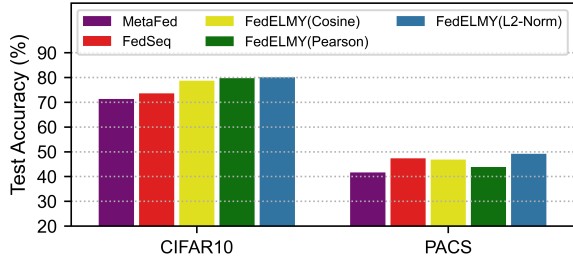

**Figure 8: Test accuracy comparison with different diversity control measures, where FedELMY (L2-Norm) means we will use the L2-norm (Euclidean) distance as our diversity control measure to train the models.**

shown in Fig. 7. The optimal combination of these parameters was then employed in subsequent training. It is evident from the figure that regardless of the adopted search strategies, the performance robustness of our method remains relatively undisturbed across all four examined parameters. This evidence suggests that the performance of our method is robust to the change of hyperparameters.

*4.4.3 Client order.* To explore the influence of the order in which domains are trained in domain-shift tasks, we applied our method to the PACS dataset with various domain training orders, as depicted in Table 4. As we can see, irrespective of the order chosen for domain training, our method consistently surpasses the performance of the baseline methods. This signifies our method's robustness in handling diverse domain order training.

*4.4.4 Diversity control measures.* Fig. 8 presents the impact of various diversity control measures applied with different distances. Notably, the L2-norm emerges as the best measure, yet, the remaining measures also exceed the performance of baseline methods in most cases. This superiority of the L2-norm is attributed to its ability to measure the true distance between points directly and precisely, offering a more sensitive and effective constraint on point dispersion in the space, thereby yielding superior results.

### 4.5 Case study

In this section, we visualize the pair-wise distances of the models in the model pool to validate the effectiveness and authenticity of our method. For more case studies, such as the visualization of the classification results of different methods, please refer to our supplementary materials.

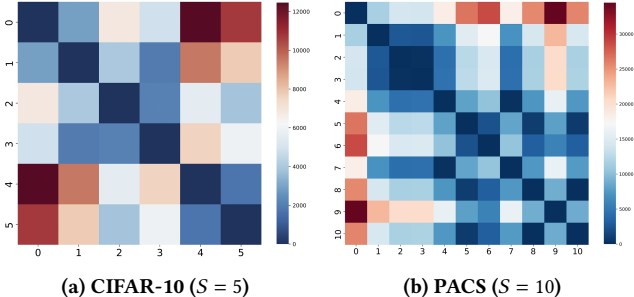

**(a) CIFAR-10 ($S = 5$)**   **(b) PACS ($S = 10$)**

**Figure 9: Heatmap of pairwise L2-norm (Euclidean) distances of all trained models within the final client's model pool $\mathcal{M}^N$ with size $S + 1$.**

Fig. 9 illustrates the L2-norm distance matrix for each pair of models within the final client's model pool $\mathcal{M}^N$, after the completion of training. It is evident that the pairwise distances among all trained models in the model pool display a remarkable variation, without an apparent correlation or trend, such as monotonically increasing/decreasing. This confirms the significant diversity among the models within the pool, attesting to the efficacy of our method in fostering model diversity. By strategically enhancing the disparities between models, we can substantively improve the performance of our approach.

For more experimental results, such as the performance of our method on more clients (up to 100 clients), and different data distribution (like *Dir(0.1)*), more grid search results for the PACS dataset, the impact of different model structures, and PFL adaptation, please refer to our supplementary materials.

### 5 Conclusion

In this paper, we highlight the significance of one-shot sequential federated learning for alleviating the communication burdens of current collaborative machine learning paradigms and address the challenge posed by non-IID data. We present a novel one-shot SFL framework with the local model diversity enhancement strategy to reduce communication costs and effectively improve the global model. In particular, we design the local model pool with two regularization terms as a diversity-enhanced mechanism to improve model performance and mitigate the effect of non-IID data. The effectiveness of our method was demonstrated with superior performance on extensive experiments across several datasets including both label-skew and domain-shift tasks. In the future, we will consider integrating more advanced privacy protection measures, adapting to more federated learning settings, and dealing with real-time data under online learning environments to further enhance the feasibility and scalability of the proposed framework.

### 6 ACKNOWLEDGMENTS

This research/project is supported by the National Research Foundation Singapore and DSO National Laboratories under the AI Singapore Programme (AISG Award No: AISG2-RP-2020-018).

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
