# OpenReview forum: "One-Shot Sequential Federated Learning for Non-IID Data by Enhancing Local Model Diversity"
_acmmm.org/ACMMM/2024/Conference — MM2024 Poster_

### Official Review · Reviewer_r3S1 · 2024-04-28

**Rating:** 3
**Confidence:** 2

**Summary:**

The paper talks about the  challenges in federated learning (FL), specifically focusing on the issues associated with non-IID data distributions in one-shot and sequential federated learning (SFL) paradigms. Traditional federated learning often relies on parallel settings (PFL), which can incur substantial communication and computational costs. To mitigate these costs, one-shot and SFL have been introduced as innovative approaches. However, the non-IID nature of data across clients remains a critical hurdle, limiting the effectiveness of these models due to minimal communication among the clients.

The paper introduces a novel framework called FedELMY (Federated Learning by Enhancing Local Model diversitY), which significantly improves the performance of one-shot sequential federated learning for non-IID data. The key innovation of FedELMY is the enhancement of local model diversity through the creation of a local model pool for each client. This pool includes diverse models generated during local training, which not only captures a broader spectrum of the data distribution but also incorporates two distinct distance measurements to further enhance model diversity and mitigate the effects of non-IID data.

**Strengths:**

1. Paper Organization:
The paper is well-organized, effectively facilitating a clear and intuitive understanding of the authors' ideas. This structure significantly aids in comprehending the proposed methodologies and their implications.

2. Experimental Setup and Comparisons:
The experimental setup is detailed and the comparisons are thorough. These elements are crucial for validating the effectiveness of the proposed approach and provide a solid foundation for the study's conclusions.

3. Additional Discussions and Code Descriptions:
I appreciate the further discussions and detailed code explanations included in the appendices. These additions are invaluable for replicating the experiments and understanding the finer details of the implementation.

4. Central Idea of the Paper:
The central idea of the paper is both straightforward and intriguing. It proposes generating a range of diverse models to effectively address the challenges posed by Non-IID Data in Sequential Federated Learning. This innovative approach has the potential to make a contribution to the field.

**Limitations:**

Thank you very much for your submission and for trusting the multimedia community. The following contents are not so much limitations of the paper as points for discussion.

1. **Curiosity about the choice of distance parameters d1 and d2**: What is the reason for choosing the L2 distance for the distance parameters d1 and d2 in this paper? Why not consider other types of distance metrics?

2. **Regarding the setting of the hyperparameter $\alpha$**: If I have overlooked something, please remind me. I do not seem to have seen the settings for the hyperparameter $\alpha$ mentioned in the paper, as the section 4.1 Experimental Setup only seems to mention $\beta = 0.5$.

3. **Interest in the settings of the two hyperparameters**: The settings of these two hyperparameters are interesting because they respectively require the model to distance itself from previous models and to stay close to the initial model. Therefore, I speculate that one of the constraints might become ineffective depending on the settings of these parameters. However, I am puzzled by the conclusions in Figure 7(c) and Figure 7(d). If this model is not sensitive to these two hyperparameters, does it imply that setting them to zero would not significantly impact the performance of the algorithm?

4. **Regarding Table 3**: It appears that the performance improvement of the algorithm mainly comes from the model pool (about a 5-point increase), while the contributions of d1 and d2 seem quite limited (around 1%). Furthermore, considering Figure 7, when $\alpha$ is set to less than 0.06 and $\beta$ to 0.6 or 0.8, the performance of the algorithm appears to be inferior to just using the model pool version. The analysis in Figure 7 indicates that the algorithm is robust to the performance variations with $\alpha$ and $\beta$, but it seems that different settings can even lead to negative outcomes.

In summary, I find the idea of the paper interesting, but I am unable to determine the effectiveness of d1 and d2 through either theoretical reasoning or experimental results. Therefore, I believe this paper might require further discussion.

**Suitability:**

2

---

### Official Review · Reviewer_qaVd · 2024-05-08

**Rating:** 4
**Confidence:** 4

**Summary:**

This paper proposed a novel one-shot sequential federated learning training method, called FedELMY, which is designed to solve the non-IID problem. The proposed FedELMY adopts a local model diversity-enhancing strategy for improving model performance. Specifically, FedELMY requires each client to train a model pool for diversity and leverage the averaged model of the model pool as the training output of the client for performance improvement.
Besides, they introduce two distance regularization terms to enhance diversity and mitigate the negative impact of non-IID data. Experimental results have shown superior performance compared with existing baselines.

**Strengths:**

- The paper as a whole is well-organized, and the figures and graphs throughout the paper are elaborate and clear.
-  A novel framework to improve the global model performance in one-shot sequential federated learning.
- The idea of using the diversity strategy to improve the performance is novel.
- The experimental results are comprehensive and well-grounded.

**Limitations:**

- The topic/motivation of this paper needs to be further polished. In my understanding, this paper focuses on improving the performance of the one-shot sequential federated learning by introducing the diversity strategy. It seems like solving a common problem in a specific area. The non-IID is a common problem in FL, and the paper tries to solve the problem in the new proposed one-shot and sequential setting. It is important to illustrate how hard the non-IID problem is in this new setting, compared with traditional parallel settings, one-shot settings, and sequential settings. In my opinion, the traditional parallel setting and the one-shot setting are sensitive to the non-IID data distribution, while the non-IID has less impact on the sequential setting which can be supported in Table 1 in this manuscript. So, what is the new challenge in solving the non-IID problem by combining the one-shot and sequential? Besides, the main contribution of this paper might be improving the performance of the global model, rather than solving the non-IID problem, since only a regularization term is designed for this. I consider the reason for the performance degradation may lie in introducing the one-shot setting rather than non-IID.

- Motivation aside, the paper tries to improve the performance by introducing the diversity strategy. They design a model pool for diversity, however demanding a higher computational cost for each client. The computation cost brought by the designed FedELMY is important. The paper should provide more detail about the relationship between the computation cost of building the model pool (varying S, rather than local epoch or the number of clients N) and the performance gain.
- The objective of diversity and mitigating non-IID might be contrasted. As illustrated in Figure 2, the d1 (diversity) and d2 (non-IID impact). To enhance the generalization, averaging a more diverse model is better, which requires a larger d1. However, the significant negative impact brought by non-IID is the local model diversity, that is the d2 should be limited. As designed in Eq.9, I consider the relationship between these two distances (hyperparameters \alpha and \beta) to be the key factor of the performance, which needs to be further explored.

**Suitability:**

2

---

### Official Review · Reviewer_dr1f · 2024-05-24

**Rating:** 3
**Confidence:** 3

**Summary:**

The paper addresses the topic of one-shot federated learning within a sequential federated learning framework. It introduces FedELMY algorithm, which incorporates a Model Pool to tackle the issue of data heterogeneity across clients. Additionally, it introduces a loss function designed to measure the heterogeneity of model parameters within the model pool. The focus on one-shot federated learning is  important as it can significantly enhance communication efficiency. Therefore, I believe the work in this paper is well-motivated. According to the comparison results presented in the paper, the proposed FedELMY algorithm outperforms other baseline methods.

**Strengths:**

1. The basic idea of the proposed algorithm is easily understandable and intuitive. I especially appreciate the concept of the model pool with the punishing term in the loss function. To my knowledge, this idea is novel and interesting.
2. The paper features great figures and tables, particularly Figure 3 and Algorithm 1, which significantly enhance its readability.
3. The experiments are well-designed, and the results are clearly presented.
4. The availability of open-source code enhances the reproducibility of the experimental results.

**Limitations:**

Here are my concerns or suggestions:
1. Some important content in the section of introduction is described too generally:

    1a) Lines 104-106: "Combining multiple networks" is the basic idea of federated learning. This might imply something beyond just the aggregation of models, which should be clarified. The paper uses "networks" to refer to both communication networks and neural networks, which can cause confusion.

    1b) Lines 108-110: It is unclear why applying such a diversity strategy in SFL is not feasible. More explanation is needed.

2. One of the core parts of the paper is using the proposed loss function in Equation (9) to address the impact of non-IID data. However, there is no theoretical analysis of this new loss function. I recommend using the convergence analysis approach from FedProx, which also integrates a term for dissimilarity in the loss. Without theoretical analysis, it is difficult to understand how hyperparameters affect convergence performance, relying only on experiments with various alpha and beta values.

3. For label skew, it is suggested to use the Dirichlet distribution and validate the performance impact at different levels of non-IID data in terms of label skew.

4. If there are fewer than 10 clients, the experiments may not be convincing, unless it is only designed for small number of clients.

5. When the number of local epochs is equal to 100 or 200, there might be overfitting on the local dataset. This should be clarified.

6. Despite the computational cost, the locally required memory could be a significant issue. Later-selected clients need to cache too many models, which may be unrealistic when the number of clients is high.

7. Formulas (5) and (6) are written in different styles, which should be changed for consistency.

**Suitability:**

3

---

### Meta-Review · Program_Chairs · 2024-07-01

**Recommendation:** Accept (Poster)
**Confidence:** 4

**Metareview:**

This paper addresses the topic of one-shot federated learning within a sequential federated learning framework. Not all reviewers hold positive stands for this paper.

In my stand, my major concerns include:

- lack of theoretical analysis for the proposed method, like convergence analysis. For federated learning, such analysis is important, and only empirical research is not enough.

- I am not convinced by the experimental setup, which considers only 10 and 4 clients for different datasets. A more commonly used setup is 100 clients in most existing federated learning frameworks.

*** TPC Addendum ***
This paper attracted a diversity of opinions. Given the topic, positive average scores, and increase in scores after rebuttal, the TPC recommends that the conversation continue at the conference with a poster presentation.